# Zero-Shot Medical Image Segmentation Based on Sparse Prompt Using Finetuned SAM

**Tal Shaharabany**              SHAHARABANY@MAIL.TAU.AC.IL
**Lior Wolf**                   WOLF@CS.TAU.AC.IL
*Tel Aviv University, Israel*

**Editors:** Accepted for publication at MIDL 2024

## Abstract

Segmentation of medical images plays a critical role in various clinical applications, facilitating precise diagnosis, treatment planning, and disease monitoring. However, the scarcity of annotated data poses a significant challenge for training deep learning models in the medical imaging domain. In this paper, we propose a novel approach for minimally-guided zero-shot segmentation of medical images using the Segment Anything Model (SAM), originally trained on natural images. The method leverages SAM's ability to segment arbitrary objects in natural scenes and adapts it to the medical domain without the need for labeled medical data, except for a few foreground and background points on the test image itself. To this end, we introduce a two-stage process, involving the extraction of an initial mask from self-similarity maps and test-time fine-tuning of SAM. We run experiments on diverse medical imaging datasets, including AMOS22, MoNuSeg and the Gland segmentation (GlaS) challenge, and demonstrate the effectiveness of our approach. Our code is publicly available at https://github.com/talshaharabany/ZeroShotSAM

## 1. Introduction

Manual segmentation of regions of interest in medical images by experts is time-consuming and prone to inter-observer variability. Therefore, the development of automated segmentation algorithms has become increasingly important. Indeed, automated medical image segmentation plays a crucial role in diagnosis (Devunooru et al., 2021), treatment planning (Sharma and Aggarwal, 2010), and screening (Norman et al., 2018).

Deep learning techniques, particularly convolutional neural networks (CNNs), have shown remarkable success in various image analysis tasks, including medical image segmentation (Wang et al., 2021; Patel et al., 2021). Recently, transformer based models have shown improved performance (Dosovitskiy et al., 2020). Such deep models can learn complex patterns and features from large-scale annotated datasets, enabling them to achieve state-of-the-art performance in segmentation tasks. However, the success of deep learning methods heavily relies on the availability of annotated training data, which is often limited and expensive to acquire in the medical domain.

Zero-shot segmentation offers a promising solution to address the challenges associated with data scarcity in medical imaging. Unlike traditional supervised learning approaches, zero-shot segmentation aims to perform segmentation tasks without relying on annotated data from the target domain. Instead, it leverages pre-trained models trained on related domains or tasks and generalize to a test image from an unseen domain. This approach has

the potential to significantly reduce the annotation burden and accelerate the development of segmentation solutions in medical imaging.

In this paper, we propose a novel method for zero-shot segmentation in medical images by adapting a deep learning architecture originally trained on natural images. Specifically, we rely on the seminal SAM foundation model by (Kirillov et al., 2023).

Our approach aims to bridge the gap between natural and medical image domains, enabling accurate and robust segmentation of anatomical structures and pathological regions without the need for extensive labeled medical data. It does require a sparse prompt in the form of a handful of forground and background points that are marked on the test image.

We demonstrate the effectiveness of our method through comprehensive experiments on diverse medical imaging datasets. As we show, while SAM itself has zero-shot capabilities, those obtained by our method far surpass these baseline capabilities.

## 2. Related Works

In the domain of medical imaging, segmentation plays a crucial role. Numerous deep learning-based approaches have been proposed for medical image segmentation (Zhou et al., 2018; Xiao et al., 2018; Wang et al., 2021; Patel et al., 2021; Shaharabany and Wolf, 2022). Utilizing convolutional neural networks (CNNs), these methods automatically delineate anatomical structures and pathological regions from medical images. While these algorithms offer substantial improvements in accuracy and efficiency compared to manual segmentation, they necessitate a significant amount of annotated data. Our solution addresses this challenge by harnessing the power of large pretrained "foundation" models, allowing for effective segmentation without reliance on labeled data.

The Segment Anything Model (SAM) has emerged as a powerful tool for segmentation tasks in computer vision. Developed by Kirillov et al. (2023), SAM introduces a versatile architecture capable of segmenting arbitrary objects in natural images. SAM utilizes a combination of sparse and dense prompts to guide the segmentation process, enabling accurate delineation of objects of interest. Several studies have explored the effectiveness of SAM in medical domain under fully-supervised settings tasks (Shaharabany et al., 2023; Wu et al., 2023; Xie et al., 2024; Ma et al., 2024), demonstrating its ability to outperform traditional methods in segmenting complex scenes. Utilizing SAM, our method operates under zero-shot conditions, indicating the absence of labeled data and training sets. (Shi et al., 2023; Mattjie et al., 2023; Roy et al., 2023) employ SAM also in zero-shot settings, without engaging in weight fine-tuning, while our methodology involves fine-tuning of the SAM image encoder based on its self-similarity map.

Learning to distinguish between the foreground and background of medical imaging with few marked points, positive and negative, can be used with a contrastive loss (Zhai et al., 2023) or, as we do, employing transfer learning from a method that already was trained on such a prompt (Xie et al., 2024). Our approach builds upon the sparse prompt encoder of SAM, but unlike previous work, achieves this in the zero-shot setting, without performing fine-tuning on the training set.

Zero-shot learning has emerged as a promising paradigm for addressing the challenge of data scarcity by leveraging knowledge transfer from related domains (Bucher et al., 2019; Narayan et al., 2020; Tewel et al., 2022; Mahapatra et al., 2021), especially zero-shot

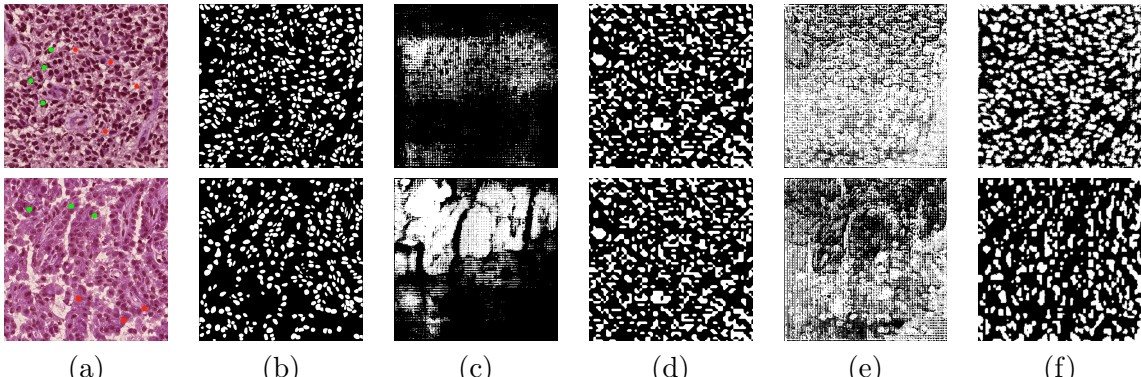

(a)          (b)          (c)          (d)          (e)          (f)

Figure 1: Sample results of segmentation of an image from the Monu dataset. (a) the input image, with marked positive and negative points. (b) the ground truth mask. (c) SAM output mask with the points as prompt. (d) The initial mask $J$ extracted from the self-similarity of SAM's encoder. (e) SAM output mask with mask prompt using $J$. (f) our result. The first row employs 4 random positive and 4 random negative points. The second employs 3 and 3. The encoder used is Vit-B.

segmentation methods (Bian et al., 2021; Ma et al., 2021) which offer a cost-effective and efficient solution for medical image segmentation, paving the way for automated analysis of medical images without the need for extensive annotation efforts. These previous work employ a different setting and cannot be compared directly with our results.

Recent work have used self-similarity maps for unsupervised and weakly supervised localization tasks for natural images (Siméoni et al., 2021; Wang et al., 2023). These identify a pattern that is well supported at multiple spatial locations in the latent space. Our approach is based on a two-stage solution, in the first stage we extract an initial mask from the self-similarity maps based on the locations provided in the sparse prompt directly.

Friebel et al. (2022); Berg et al. (2019); Pachitariu and Stringer (2022) and others integrate human-machine interactions to accelerate the annotation process for medical objects. These methods typically leverage pretrained models from medical domains in conjunction with label sketches. In a manner akin to our methodology, these techniques can be customized for utilization in a single-image context, employing point-prompt based strategies without requiring further adjustments or refinement. In order to compare with such approaches, we ignore the utilization with pretrained models from medical domains and compare with a baseline that incorporates two components that are frequent in these methods: a superpixel technique and color clustering.

## 3. Method

In this section, we describe the application of the Segment Anything Model (SAM) (Kirillov et al., 2023) to perform zero-shot segmentation on medical images, which is an out-of-distribution domain. Our method consists of two stages (i) the extraction of initial mask from self-similarity maps, and (ii) a test-time image-specific fine-tuning of SAM.

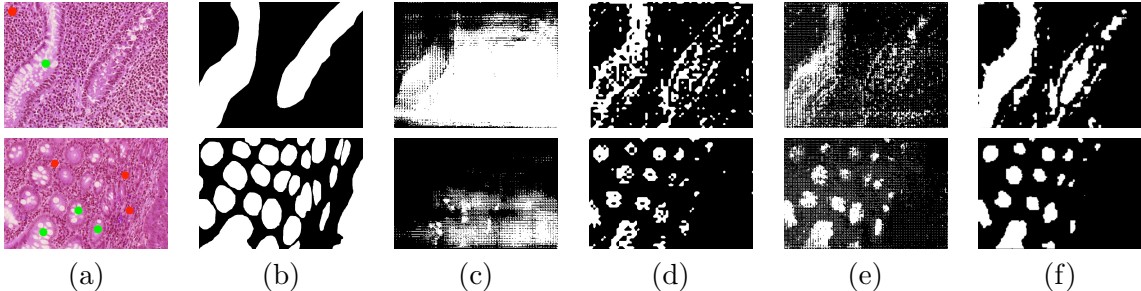

(a)  (b)  (c)  (d)  (e)  (f)

Figure 2: Segmenting images from the Glas dataset. (a) the input, with marked positive and negative points. (b) the ground truth mask. (c) SAM output mask with the points as prompt. (d) the initial mask $J$. (e) SAM output mask with mask prompt using $J$. (f) our result. The 1st row employs 1 positive and 0 negative points; the last has 3 each. ViT-Base encoder is used.

The Segment Anything Model $S$ generates an output mask $M \in R^{H \times W}$ given an RGB input image $I \in R^{H \times W \times 3}$ (for gray scale medical images we duplicate the image across the three channels) and two types of prompts: (i) a sparse prompts $P_s$ which related to bounding boxes $B \in R^4$ or two sets of positive and negative points $P \in R^2$, and (ii) a dense prompts $P_d \in R^{W \times H}$ which is an estimated segmentation map. The output map is obtained as $M_o = S(M, P_s, P_d)$.

SAM consists of three main components. The first component is the visual encoder $E_i$, which is based on a visual transformer (ViT) (Dosovitskiy et al., 2020) with a patch size of $16 \times 16$ and one of three architectures (i) "Base" with 12 transformers layers, (ii) "Large" with 24 transformers layers, or (iii) "Huge" with 32 transformers layers. The encoder maps the input image $I$ to a tensor $Z_i \in R^{d_i \times \frac{W}{16} \times \frac{H}{16}}$, where $d_i$ is 768.

The second component of SAM is the two prompt encoders: one for the sparse prompt $P_s$ and one for the dense prompt $P_d$. These encoders $E_s, E_d$ map the corresponding input prompt to tensors $Z_s \in R^{d_s \times \frac{W}{16} \times \frac{H}{16}}$ and $Z_d \in R^{d_s \times \frac{W}{16} \times \frac{H}{16}}$, respectively. For each encoder, there is a null prompt for not using this specific prompt.

The last component of SAM is the mask decoder $D_m$, which receives the visual representation $Z_i$ together with the prompts representations $Z_s$ and $Z_d$ and generates an outputs mask $M_o$. In the setting of the zero-shot medical segmentation, the model receives the image and several points that indicate foreground $P_{pos}$ and background $P_{neg}$ pixels.

The straightforward option is to use SAM's sparse prompt encoder $E_s$ with $P_s$ that is the concatenation of the positive and the negative points, together with the original encoder $E_i$ in order to get the output mask from $D_m$. However, since the data is out-of-distribution, the output mask is inaccurate and requires further refinement.

We, therefore, propose a two-stage method that first extracts an initial mask $J$ from the visual embeddings $Z_i$ returned by $E_i$, and then applies test time fine-tuning of SAM's image encoder to bring SAM's output mask closer $J$. The other encoders and the decoder are not tuned (frozen networks).

**First stage - From Self Similarity Maps to Initial Mask**   We consider the normalized visual latent space as $Z_i$ and calculate the cosine similarity between the vectors in $\mathbb{R}^{d_i}$ in each pair of spatial locations. The output similarity map $A \in R^{c_w c_h \times c_w c_h}$, where $c_h$ and $c_w$ are $\frac{H}{16}$ and $\frac{W}{16}$, respectively. For SAM cases, the input image size is fixed to $1024 \times 1024$, so $c_w$ and $c_h$ equals to 64.

The map $A$ has a diagonal equal to 1 and $A_{uv}$ stands for the cosine similarity between $Z_i(\lfloor \frac{u}{c_h} \rfloor, u \mod c_h)$ and $Z_i(\lfloor \frac{v}{c_w} \rfloor, v \mod c_w)$, where indexing into $Z_i$ by spatial dimensions and provide a vector in $d_i$.

For each input point $(x, y)$, whether it $P_{pos}$ or $P_{neg}$, we calculate the corresponding latent pixel by dividing the pixel values by 16, and sampling the $u = x * 64 + y$ row of $A$. The obtained row is denoted as $A(u)$ and can be viewed as a spatial map of size $c_w \times c_h$.

We then define two maps: a positive maps $M_{pos}$, and a negative map $M_{neg}$ (both contain only positive numbers) by a simple summation: $M_{pos} = \sum_{(x,y)\in P_{pos}} A(x * 64 + y), \quad M_{neg} = \sum_{(x,y)\in P_{neg}} A(x * 64 + y)$

In the case in which no negative points are used, the algorithm normalizes the value of $M_{pos}$ by subtracting the minimum value in $M_{pos}$ from each pixel in $M_{pos}$ and dividing by a $d = \max M_{pos} - \min M_{pos}$. Then, the initial mask $J$ is given by thresholding the obtained normalized map at a value of 0.5.

In the case of mixed label point, the algorithm obtains the initial mask $J$ by comparing each pixel in the two maps, i.e., $J(x, y) = \frac{1}{2}(\text{sign}(M_{pos}(x, y) - M_{neg}(x, y)) + 1)$. A visualization of $J$ is presented in App. A.

**Second Stage - Test time Optimization of SAM**   In the second stage, we utilize the initial mask $J$, which indicates foreground and background pixels, and finetune SAM's visual encoder $E_i$ while the decoder weights are frozen and both prompts are null, by considering $M_o = S(I, null, null)$. During the optimization, for each test image separately, the algorithm initializes $S$ with the pre-trained weights and optimizes for that single test image independently of any other test images.

The following loss terms are used during this fine-tuning process $L(I) = L_{BCE}(M_o, J) + L_{dice}(M_o, J)$, where the BCE loss is defined as $L_{BCE}(X, Y) = -Y * log(X) - (1 - Y) * log(X)$, the Soft-Dice loss is defined as: $\mathcal{L}_{dice}(X, Y) = 1 - \frac{(2 * \sum_{ij} X_{ij} * Y_{ij}) + 1}{2 * \sum_{ij} X_{ij} * Y_{ij} + \sum_{ij}(1 - X_{ij}) * Y_{ij} + \sum_{ij} X_{ij} * (1 - Y_{ij}) + 1}$

## 4. Experiments

In this section, we detail the experimental setup used to validate the efficacy of our proposed zero-shot sparse-prompt segmentation method employing the Segment Anything Model (SAM) (Kirillov et al., 2023). Our study is structured around two primary stages: (i) the extraction of the initial mask $J$ from self-similarity maps and (ii) the test-time fine-tuning of SAM. We, therefore, also provide multiple alternative results, which apply only some of these elements: the SAM model with the point prompt, SAM model on top of the initial mask $J$, and the initial mask itself.

To assess the effectiveness of our network in image-based segmentation tasks, we utilized the commonly employed metrics: mean Intersection-over-Union (IoU) and the Dice-Score.

Three datasets are used: **AMOS22** (Ji et al., 2022) Abdominal CT Organ Segmentation with 240 test images and 14 different categories. We adopted the 2D slicing configuration

Table 1: Zero-shot segmentation: dice scores for AMOS22 dataset with a varying number of positive points. SAM w/P utilizes the positive points as sparse prompts. The supervised state-of-the-art nnUNet-3D (Isensee et al., 2021) serves as the upper limit for comparison with the method.

| | Spl. | R.Kid. | L.Kid. | GallBl. | Esoph. | Liver | Stom. | Aorta | Postc. | Pancr. | R.AG. | L.AG. | Duod. | Blad. |
|---|---|---|---|---|---|---|---|---|---|---|---|---|---|---|
| Super-Clustring w/P=1 | 0.329 | 0.454 | 0.474 | 0.311 | 0.287 | 0.272 | 0.212 | 0.422 | 0.358 | 0.220 | 0.038 | 0.023 | 0.235 | 0.245 |
| SAM w/P=1 | 0.632 | 0.759 | 0.770 | 0.616 | **0.382** | **0.577** | 0.508 | 0.720 | 0.453 | **0.317** | 0.085 | 0.196 | 0.339 | 0.542 |
| J w/P=1 | 0.512 | 0.629 | 0.701 | 0.576 | 0.292 | 0.467 | 0.428 | 0.680 | 0.413 | 0.281 | **0.150** | **0.260** | 0.349 | 0.532 |
| Ours w/P=1 | **0.667** | **0.781** | **0.784** | **0.686** | 0.377 | 0.533 | **0.528** | **0.767** | **0.463** | 0.309 | 0.135 | 0.182 | **0.358** | **0.611** |
| Super-Clustring w/P=3 | 0.430 | 0.488 | 0.482 | 0.377 | 0.293 | 0.363 | 0.274 | 0.455 | 0.398 | 0.248 | 0.044 | 0.039 | 0.272 | 0.298 |
| SAM w/P=3 | 0.733 | 0.784 | 0.786 | 0.683 | 0.448 | **0.658** | **0.577** | 0.758 | 0.493 | 0.343 | 0.129 | 0.240 | 0.325 | 0.631 |
| J w/P=3 | 0.594 | 0.663 | 0.749 | 0.655 | 0.378 | 0.535 | 0.502 | 0.727 | 0.462 | 0.328 | 0.201 | **0.272** | 0.332 | 0.603 |
| Ours w/P=3 | **0.742** | **0.804** | **0.797** | **0.713** | **0.468** | 0.638 | 0.557 | **0.798** | **0.502** | **0.352** | **0.214** | 0.254 | **0.365** | **0.692** |
| Super-Clustring w/P=10 | 0.654 | 0.652 | 0.661 | 0.596 | 0.548 | 0.615 | 0.661 | 0.638 | 0.533 | 0.445 | 0.303 | 0.411 | 0.385 | 0.594 |
| SAM w/P=10 | 0.857 | 0.855 | 0.857 | 0.800 | 0.643 | 0.811 | 0.759 | **0.842** | **0.637** | 0.538 | **0.405** | 0.516 | **0.480** | 0.789 |
| J w/P=10 | 0.834 | 0.802 | 0.816 | 0.778 | 0.662 | 0.795 | 0.729 | 0.811 | 0.602 | 0.518 | 0.332 | 0.502 | 0.421 | 0.744 |
| Ours w/P=10 | **0.872** | **0.862** | **0.866** | **0.811** | **0.677** | **0.835** | **0.779** | 0.837 | 0.622 | **0.551** | 0.375 | **0.532** | 0.455 | **0.792** |
| nnUNet-3D (supervised) | 0.978 | 0.951 | 0.951 | 0.903 | 0.856 | 0.978 | 0.919 | 0.961 | 0.923 | 0.856 | 0.790 | 0.815 | 0.814 | 0.929 |

outlined by Roy et al. (2023). **The MoNuSeg dataset** (Kumar et al., 2019) contains 38 microscopic images depicting seven organs in its training set, annotated with 21,623 individual nuclei, where the test set comprises 14 similar images. **The Gland segmentation (GlaS) challenge** (Sirinukunwattana et al., 2017) comprises a curated dataset consisting of 85 high-resolution images designated for training and 80 images reserved for testing.

In the experiments, positive points are sampled randomly from coordinates where the pixel value in the ground-truth mask equals 1, while negative points are sampled from coordinates where the pixel value is 0. The random sampling means that the points are, in many cases, less typical than those a human would select as guidance. Since the sampling process introduces a variation, each experiment result is averaged over five iterations on each test image and then averaged over the entire test set. Naturally, the same points are used for comparing the various algorithms.

Inspired by previous work, we add a strong zero-shot baseline ("Super-Clustering") that combines superpixel segmentation and color clustering techniques for zero-shot segmentation. Additional details are available in App. C.

**Optimization details** During the test-time fine-tuning of $S$, we employ the ADAM optimizer with an initial learning rate of $1 \cdot 10^{-5}$. A batch size of 1 is utilized, and no further augmentation has been implemented. We conduct training on NVIDIA A5000 with 24GB GPU RAM. The number of fine-tuning iterations for $S$ network fine-tuning was set to 100. The SAM pre-trained weights that we utilized were based on the ViT 'base' and 'large' architecture, Due to resource constraints, we opt not to use the Huge ViT-based encoder, which necessitates more than 48GB GPU memory. The SAM image encoder is provided with input images of size $1024 \times 1024$. We adhere to the pre-processing protocol outlined by SAM (resize, padding, normalization) to ensure optimal performance. The test time fine-tuning operation takes 40 and 80 seconds per image for Vit-base and Vit-large respectively.

**Results** Tab. 1 presents zero-shot segmentation results for the AMOS22 (Ji et al., 2022) dataset across different numbers of positive points (P) and various methods. Notably,

Table 2: MoNu results for different number of positive points $n_{pos}$ and negative points $n_{neg}$. All methods are zero-shot methods with a handful of points on the test image used for supervision. SAM w/P uses the points as the sparse prompt. SAM w/J uses the map $J$ obtained from the first phase of our method as a dense prompt. The first number indicates the dice score and the second the IoU. Best (2nd best) results are indicated as bold (underlined). For reference, the fully supervised AutoSAM method, trained on the entire training dataset, which is not used in zero-shot at all, obtains a Dice score of 82.43 and an IoU of 70.17 (Shaharabany et al., 2023).

| $n_{pos}$ $n_{neg}$ | ViT Base | | | | ViT Large | | | | |
| | SAM w/P | SAM w/J | J | Ours | SAM w/P | SAM w/J | J | Ours | Super-Clustring |
|---|---|---|---|---|---|---|---|---|---|
| 1 0 | 11.84/6.68 | 29.62/17.52 | 31.18/19.56 | **40.43/27.81** | 9.85/5.43 | 33.37/20.20 | 25.64/15.54 | 29.47/18.80 | 37.11/23.32 |
| 1 1 | 11.65/6.88 | 34.32/21.08 | 36.17/22.84 | 37.71/24.98 | 3.05/1.82 | 34.28/20.90 | 41.29/26.44 | **46.78/31.68** | 44.77/31.67 |
| 2 0 | 10.31/5.82 | 30.40/18.05 | 33.71/20.72 | **38.86/25.32** | 10.80/6.05 | 33.61/20.38 | 31.73/19.57 | 32.75/20.92 | 37.95/23.65 |
| 3 0 | 9.05/5.10 | 31.32/18.79 | 32.26/19.71 | 37.52/24.13 | 14.00/7.88 | 33.51/20.30 | 37.93/23.87 | **40.85/26.68** | 38.55/24.11 |
| 2 2 | 20.86/12.52 | 34.77/21.32 | 37.33/23.68 | 40.91/28.07 | 5.01/2.89 | 34.02/20.68 | 37.53/23.83 | **42.11/28.26** | 17.08/12.24 |
| 4 0 | 6.90/3.87 | 31.40/18.85 | 35.88/22.16 | **40.84/26.30** | 11.76/6.60 | 33.59/20.37 | 36.34/22.43 | 39.19/24.76 | 38.43/24.17 |
| 5 0 | 9.42/5.37 | 30.79/18.36 | 33.59/20.35 | 36.23/22.40 | 12.57/6.98 | 33.85/20.56 | 36.90/22.95 | **39.20/24.88** | 38.53/24.10 |
| 3 3 | 18.80/11.31 | 32.13/19.45 | 42.28/28.10 | 42.48/28.65 | 7.34/4.46 | 34.03/20.70 | 42.91/27.88 | **49.09/33.82** | 21.81/15.81 |
| 4 4 | 15.16/9.14 | 32.74/19.86 | 46.43/30.67 | **54.98/39.49** | 11.49/6.97 | 34.86/21.33 | 40.85/26.22 | 46.69/31.69 | 3.62/2.42 |
| 5 5 | 18.13/10.82 | 33.57/20.50 | 45.14/29.43 | **52.37/36.34** | 8.25/4.92 | 34.10/20.76 | 43.76/28.44 | 50.45/34.80 | 4.61/3.41 |

our proposed method, labeled as "ours w/P=1," achieves the highest Dice scores for most anatomical structures at P=1, surpassing the average performance of super-clustering by more than 0.2 dice score, and outperforming competing methods SAM w/P=1 and J w/P=1 by 0.02 and 0.06 dice score on average, respectively. This highlights the effectiveness of our approach even with minimal positive points. As positive points increase to P=3, our method maintains superior performance, particularly evident in Dice scores for organs like the spleen, right kidney, and left kidney, outperforming SAM w/P=3 and J w/P=3. On average for P=3, our algorithm outperforms super-clustering, SAM w/P=3, and J w/P=3 by more than 0.24, 0.02, and 0.06 dice score, respectively. At P=10, our method (Ours w/P=10) sustains high performance across most structures, achieving the highest overall Dice score and surpassing super-clustering by more than 0.15 dice score on average, as well as baseline methods SAM w/P=10 and J w/P=10 by more than 0.005 and 0.04 dice score on average, respectively. Moreover, our method competes favorably with the supervised state-of-the-art represented by nnUNet 3D, demonstrating its potential for practical deployment in scenarios with limited annotated data. The improvement of $J$ during the finetune for images from AMOS22 presented in App. B.

The results from the MoNu dataset are presented in Tab. 2. It is evident that our method achieves the best performance, attaining a mean IoU of 39.49 with ViT-base, which significantly outperforms the top result obtained by SAM using $J$ in the mask prompt encoder, which achieved a mean IoU of 21.33 with ViT-base. Across most input marked point modes, both ViT architectures demonstrate superior performance compared to standard zero-shot with SAM and super-clustering, except in cases where a single positive point is sampled without a negative point for ViT-large. This discrepancy could be attributed to the inherent inaccuracies associated with relying on a single random representation in our

Table 3: GLAS results for different number of positive points $n_{pos}$ and negative points $n_{neg}$. The first number in each pair indicates the dice score and the second the IoU. See Table 1 for details. For reference, the fully supervised AutoSAM obtains the Dice score of 92.82 and an IoU of 87.08 (Shaharabany et al., 2023).

| $n_{pos}$ $n_{neg}$ | ViT Base | | | | ViT Large | | | | |
|---|---|---|---|---|---|---|---|---|---|
| | SAM w/P | SAM w/J | J | Ours | SAM w/P | SAM w/J | J | Ours | Super-Clustering |
| 1 0 | 45.31/33.48 | 52.11/36.95 | 44.00/30.00 | 56.81/42.38 | 56.60/42.60 | **65.76/50.90** | 48.35/33.82 | 51.36/36.71 | 65.53/50.75 |
| 1 1 | 26.76/19.31 | 54.52/39.46 | 63.38/48.00 | 65.45/50.28 | 5.72/3.84 | **65.74/50.91** | 63.77/48.81 | 64.88/50.21 | 31.79/24.55 |
| 2 0 | 42.42/31.04 | 58.83/43.57 | 61.65/45.93 | 65.79/50.91 | 57.64/43.37 | **65.82/50.95** | 63.66/48.19 | 65.26/50.19 | 65.28/50.43 |
| 3 0 | 44.24/32.60 | 59.31/44.08 | 61.87/46.26 | 65.10/50.18 | 57.89/43.53 | **65.89/51.06** | 63.90/48.66 | 65.32/50.43 | 64.72/49.89 |
| 2 2 | 29.92/21.60 | 57.95/42.83 | 63.33/48.70 | 64.71/50.66 | 4.50/2.84 | 65.75/50.89 | 68.45/53.91 | **69.45/55.16** | 16.15/12.66 |
| 4 0 | 43.56/31.99 | 60.33/45.14 | 64.13/48.84 | 66.05/51.15 | 57.64/43.39 | 65.97/51.14 | 64.62/49.55 | **66.02/51.30** | 65.14/50.35 |
| 5 0 | 42.75/31.56 | 60.23/45.10 | 64.32/49.08 | 65.96/51.08 | 57.52/43.28 | 65.96/51.43 | 65.25/50.05 | **66.78/51.94** | 64.15/49.44 |
| 3 3 | 27.29/19.72 | 59.31/44.20 | 65.22/50.52 | 66.10/51.82 | 3.18/2.03 | 65.82/50.97 | 69.94/55.26 | **71.27/56.98** | 9.24/7.48 |
| 4 4 | 26.99/19.32 | 59.29/44.23 | 69.43/54.54 | 70.36/55.76 | 4.07/2.55 | 65.95/51.11 | 69.80/55.05 | **70.56/56.12** | 4.01/3.41 |
| 5 5 | 23.27/16.64 | 58.90/43.98 | 68.02/53.26 | 69.32/55.09 | 4.22/2.69 | 65.84/50.98 | 72.02/57.71 | **72.82/58.84** | 2.71/2.17 |

algorithm, whereas the mask encoder of SAM effectively manages this issue. Consequently, for multi-point input scenarios, our algorithm demonstrates superior performance. As a general trend, given more points, the performance improves. However, going from four to five points of each type does not increase performance. Typically results on the Monu dataset using ViT-Base are presented in Fig. 1 for Vit-Base SAM. The masks obtained by SAM, using either the point prompt or a dense prompt that relies on the initial mask we extract are inaccurate, while our full method provides adequate segmentation maps. The results for the GlaS dataset are depicted in Tab. 3. Among these, the highest performance is achieved by ViT-large, with a mean IoU of 58.84, surpassing the best performance of the standard SAM which yields a mean IoU of 51.43, and super-clustering, which obtained mean IoU of 50.75. Our algorithm demonstrates superior performance in scenarios involving four marked points; however, its sensitivity to outliers leads to decreased performance in cases with fewer points. Notably, the point encoder's ability to handle negative points appears inadequate, suggesting potential limitations in SAM's capacity to discern foreground from background on this data. Figure 2 showcases typical results on the GlaS dataset utilizing Vit-Base SAM. Similarly to Figure 1, our solution enhances the segmentation mask in comparison to SAM's results using either prompt and also greatly improves over $J$.

## 5. Conclusions

We demonstrate that SAM is not effective in zero-shot segmentation of out-of-domain medical images with multiple foreground segments, when using the sparse prompt. However, when relying on SAM's embedding, one can extract a reasonable initial mask. SAM's output when using this initial mask as a dense prompt is not better than this mask. However, when applying a test-time fine-tuning of SAM, the obtained model is considerably better on the test image at hand. As future work we would like to study the generalization ability of these finetuned networks (a simple experiment that we did not yet run) and also train a feed-forward model using meta-learning (Lutati and Wolf, 2023).

## Acknowledgments

This research was funded by the Blavatnik CS Research Fund and a grant from the Tel Aviv University Center for AI and Data Science (TAD). Additionally, support was provided by the Ministry of Innovation, Science & Technology, Israel (1001576154), and the Michael J. Fox Foundation (MJFF-022407) (to U.A). The contribution of TS is part of a Ph.D. thesis conducted at Tel Aviv University.

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

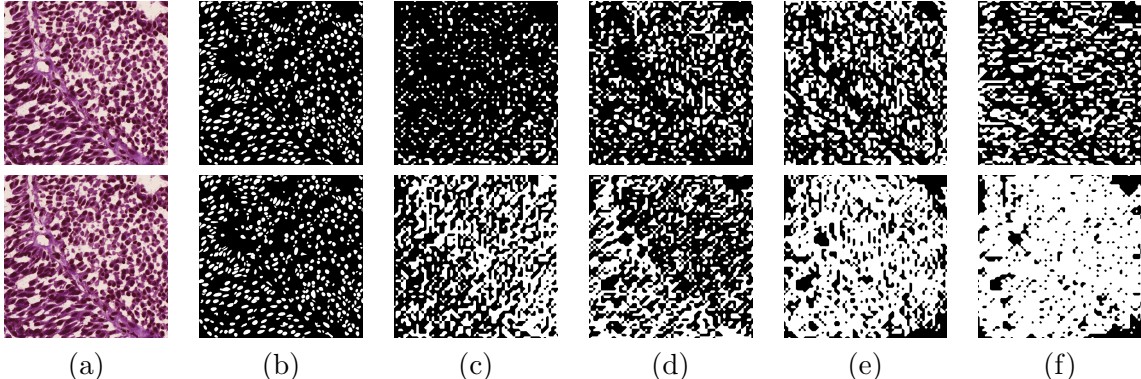

(a)       (b)       (c)       (d)       (e)       (f)

Figure 3: The initial mask $J$ for different numbers of positive points and no negative points. The first row shows successful cases, and the second failure cases, where the difference is due to the sampling of the points. (a) The input image (b) the ground truth (c) one positive point (d) two positives points (e) three positive points (f) four positive points. The SAM model is based on ViT-Large.

## Appendix A. Samples of the initial mask $J$

Figure 3 demonstrates the evolution of the self-similarity-derived mask $J$ as the number of positive points grows. Two results are presented for each case, to demonstrate the variability that arises from the random sampling process. In the typical case, more points tend to improve performance. In the failure case, where unfavorable points are sampled, more points do not lead to an improvement.

## Appendix B. Finetune Enhancement Graphs

Fig. 4 illustrates the enhancement achieved through the fine-tuning process using the initial mask $J$ for spleen segmentation, the first category, within the AMOS22 (Ji et al., 2022). The results demonstrate notable performance improvements across most points, not only refining the accuracy of the initial mask $J$ but also enhancing the segmentation for inaccurate instances.

## Appendix C. Super-Clustring

Superpixel techniques (Achanta et al., 2012) partition the image into meaningful regions based on low-level features, while color clustering groups pixels with similar color values into clusters. In our experiments, superpixels are initially generated to divide the image into homogeneous regions. Foreground super-pixels are then selected using the supervision points. We employ k-means clustering with k=2 to facilitate color clustering, effectively grouping individual super-pixels based on similar color characteristics. This combined process enhances the delineation of relevant regions in the image.

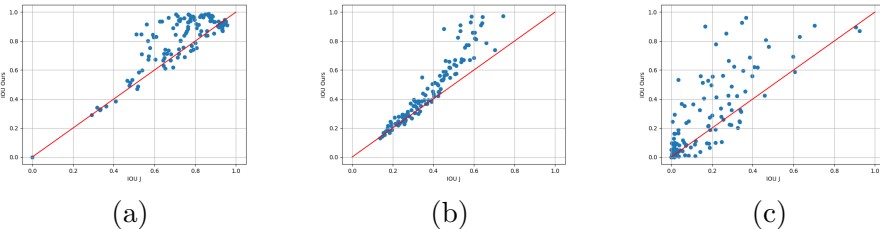

(a)        (b)        (c)

Figure 4: Comparison of segmentation performance before and after fine-tuning using the initial mask $J$ for spleen segmentation in the AMOS22 dataset. The graphs depict the enhancement observed at different levels of positive points $(P)$, specifically (a) $P = 10$, (b) $P = 3$, and (c) $P = 1$.

