# OpenReview forum: "Zero-Shot Medical Image Segmentation Based on Sparse Prompt Using Finetuned SAM"
_MIDL.io/2024/Conference — MIDL 2024 Poster_

### Official Review · Reviewer_aU6b · 2024-02-28

**Confidence:** 3
**Preliminary Rating:** 2
**Final Rating:** 3.5

**Summary:**

The authors present an approach to adapt the Segment Anything Model for zero-shot adaptation for the task of interactive / semi-automatic nuclei and gland segmentation. They present two main contributions: First, they derive a self-similarity map from point inputs, which they then use to prompt SAM, and second, they fine-tune the encoder during test-time such that the output of SAM better matches this self-similarity map. The authors report improved results compared to vanilla SAM and compared to using the self-similarity map alone on two microscopic datasets.

**Strengths:**

- The authors present an interesting approach to zero-shot, interactive segmentation, here targeted at microscopic images, which they evaluate on two image sets containing cell nuclei and glands.
- Combination of a powerful, pre-trained approach (SAM) with self-similarity (according to SAM-derived features), which seems like a very interesting approach to prompt the system in a - in some interpretation maybe - self-attention-based manner.
- Interesting approach to test-time fine-tuning which refines the representation in the encoder.

**Weaknesses:**

- Lack of comparison to other (zero-shot) approaches. It is likely that the approach is successful because the target structures are repeating patterns, which makes it difficult for single-object approaches like random walker and more complex methods that are based on that to succeed. Still, a number of pre-trained models exist that could be used for comparison. Good candidates from my perspective would be, for example, Cellpose 2.0 [1], ilastik [2], and potentially a clustering-based approach as featured in [3]. While these methods may tackle the problem from a different side or may start from different starting conditions and training dataset, from my perspective this provides important insights into whether the current approach is an exploratory experiment or actually competitive with ready-to-use tools for microscopic image analysis.
- Following this, it is unclear to what extent this approach would work also for larger objects that feature a stronger heterogeneity, e.g., organ segmentation or whether the authors present here an after all very targeted approach for a rather specific problem (which is not an issue by itself, but should be discussed).
- The literature review is rather brief on the existing studies of the (also zero-shot) capabilities of SAM for medical images. It may be interesting to further consider the by now relatively vast literature on medical image segmentation and SAM, e.g., [4,5], which have evaluated the performance of zero-shot SAM.
- The output of (vanilla) SAM looks unexpectedly bad, and I am wondering whether there could be a mistake in the implementation. Having tried out similar images with the SAM demo [6], I obtained less than perfect but rather decent nuclei segmentation masks (I understand that there may be some post-processing, but the presented segmentation maps, e.g., in Fig. 1 look completely off and also feature extremely strong checkerboard artifacts).


[1] https://www.nature.com/articles/s41592-022-01663-4
[2] https://www.nature.com/articles/s41592-019-0582-9
[3] https://academic.oup.com/bioinformatics/article/38/19/4622/6670619
[4] https://arxiv.org/abs/2304.05396
[5] https://arxiv.org/abs/2305.00109
[6] https://pubmed.ncbi.nlm.nih.gov/37296799/

**Detailed Comments:**

- Figure 1: In this figure it is rather difficult to understand false positives / false negatives; a suggestion would be to use e.g., different colors for TP, FP and FN pixels to better understand where the different approaches make mistakes.
- p. 5: "test time fine-tuning of SAM’s image encoder to bring SAM’s output mask closer J" - since J is computed from the encoder, is J static? Can the authors provide an intuition of why it is more beneficial to fine-tune the encoder instead of the the decoder?
- p. 5: I understand that the authors aim to work with a well defined similarity map; however, my feeling is that by linearizing this as proposed, the authors make this rather bulky. It may be easier to simply work here with a tensor or potentially or simply a function with two vector inputs.
- p. 6: I would be surprised if the authors use a fully thresholded version of the DICE loss as this requires a binarization of both predictions M_o and J - is it possible that the authors use a soft-Dice loss as in many other segmentation papers? (e.g., V-Net or others)
- p. 6: Details regarding the image size / input to the network etc., potential normalization, seem to be missing from the paper.
- p. 8: comparing performance results with different numbers of input samples between network settings (e.g., for MoNu 4 positive / 4 negative for the proposed method vs. 2 pos / 2 neg for SAM w/ J) is not really informative from my perspective as this information would not be available for a user in a zero-shot setting.


Spelling, typos and minor errors:
- p. 2: "effectiveness of SAM in Medical domain" - "the" missing/capitalization
- p. 2: "Learning [...] can be use with contrastive loss" - wording
- p. 4: "[...], Where di is 768" - typo
- p. 5: "The map A [...] , where indexing into Zi is by the spatial dimensions and provide a vector in di ." - I am not completely clear what you aim to state here.
- Tab. 2: "ahe Dice" - typo

**Justification Of Final Rating:**

The authors responded in detail to most of questions, and spend substantial effort during the rebuttal to add additional comparisons to the paper. I am therefore happy to increase my rating to some extent. Some concerns remain, i.e., with regard to the drop in performance for some comparison approaches and the performance of vanilla SAM. As it still may be interesting to discuss these aspects at MIDL, I'm assigning a borderline accept rating to this paper.

**Justification Of The Preliminary Rating:**

From my perspective, the paper presents an interesting approach, but is currently lacking comparisons to other zero-shot / Test-time fine tuning approaches. While I don't necessarily expect that this approach has to reach SOTA in comparison with them, it would be good to understand how large the gap is to understand whether this is a truly interesting way in the direction of zero-shot learning or whether this is just along the lines of "trying something with SAM".
Furthermore, the results of vanilla SAM look erroneous, and I would like to better understand why this is the case.

Therefore, I would like to better understand the reasoning why no comparison approaches where included and whether I am potentially misunderstanding something, which I hope the authors will answer in their rebuttal.

**Questions To Address In The Rebuttal:**

- What is the reason that the segmentation maps for SAM look so unexpectedly bad? Could there be any issues with the normalization?
- Why did the authors not compare to any existing unsupervised or zero-shot approach? What would be the result, e.g., with CellPose (unsupervised/zero-shot) or ilastik (zero-shot)?
- To what extent do the authors believe that their method is only suitable for texture-like segmentation tasks with very strong repeating patterns?

---

> ### Author Response · Authors · 2024-03-16
>
> We thank the reviewer for the comprehensive feedback.
>
> > Lack of comparison to other zero-shot approaches
>
> We recognize the significance of benchmarking against established methodologies in the field. While the setting of several of the existing existing references [1, 2, 3] may not directly align with our approach, we have added a baseline method with the specified components (superpixels and color clustering).
>
> > Larger objects
>
> Following the review, we now benchmark our method also on the AMOS22 dataset, which encompasses 14 distinct categories featuring varied sizes of medical objects. The results on the Amos22 dataset serve as compelling evidence that our method extends beyond texture-like segmentation tasks with strong repeating patterns. It also shows that the method performs well across a diverse range of medical objects with varying sizes and complexities.
>
> > The output of (vanilla) SAM
>
> We appreciate the reviewer for bringing this to our attention. The segmentation maps for SAM appear unexpectedly poor because SAM was not trained on medical data or designed for medical image segmentation tasks. Upon further investigation, we did not identify issues within our framework that may have exacerbated this problem. We note that another contribution (SAM.MD; [4]) obtained similar findings. It is worth noting that in the Meta demo, the initial SAM utilized was ViT-Huge (while we employed Large and Base versions), and additional post-processing was applied to the output mask, potentially leading to slightly improved results.
>
> > Additional literature review
>
>
> In Table 1, we compare our performances with findings from SAM.MD[4] which similarly employs SAM in zero-shot scenarios (the baseline results for SAM with the points prompt is exactly what was done in [4]). As part of our revision, we have also updated the related work section.
>
> > Fine-tuning of SAM
>
> We chose not to optimize the decoder because doing so could cause it to simply follow $J$ and converge to it without refining it. By freezing the decoder and solely optimizing the encoder, the output mask is not able to precisely follow $J$ and therefore refines it, as demonstrated in Appendix B.

---

> > ### Comment · Reviewer_aU6b · 2024-03-21
> > **Reply to author comments**
> >
> > I would like to thank the authors for their detailed comments and providing the revised version of their paper, the additional experiments and explanations. It is clear that the authors spend a lot of effort on this revision.
> >
> > I would like to briefly reply to the points mentioned by the authors:
> > > Lack of comparison to other zero-shot approaches
> >
> > I appreciate the additional superpixel / clustering baseline, but see two issues:
> > 1) I disagree with the general statement that the suggested methods do not align with the proposed setting. While the motivation of the approaches is broader, both cell-pose and ilastik can be used in a single-image, point-prompt based approach without further adaptation / selection. This request should not prompt the authors to adapt their comparison method, but from my perspective, this is not suitable represented in the related work section.
> > 2) the selected superpixel approach seems to tremendously deteriorate when even a single negative point is provided (similarly actually vanilla SAM). Do the authors have an explanation for that?
> >
> > > Larger objects
> >
> > I would like to thank the authors for adding this dataset. Do the authors have an explanation why SAM + J performs considerably worse compared to SAM alone (in contrast to the other datasets)?
> >
> > > The output of (vanilla) SAM
> >
> > I thank the authors for the additional check within their own data. After re-reading the SAM.MD short paper, I did not find any comment on this or comparable artifacts. Could the authors point me to this statement?
> >
> > > Fine-tuning of SAM
> >
> > Thanks, that makes it more clear!
> >
> > > Renaming to soft-Dice loss:
> >
> > It seems that the authors may have misunderstood my comment. I don't really have an issue with the name (which is often called "Dice loss" but with the equation on pg. 5, which uses binarized TP / FP / FN. Since this makes gradient computation very sparse, I kindly ask the authors to revisit this equation.
> >
> > I completely understand where this comes from, but small font sizes in the tables and potentially changes in the vspacing makes the paper rather cramped. The authors may want to consider moving some tables / part of the tables also into the supplementary material.

---

> ### Author Response · Authors · 2024-03-22
>
> We truly appreciate the detailed reply and are thankful for the ongoing dialog.
>
> > Lack of comparison to other zero-shot approaches
>
> We appreciate the feedback and acknowledge the valid point raised regarding the alignment of the suggested methods with the proposed setting. Upon reconsideration, we agree that both CellPose and Ilastik can indeed be adapted for use in a single-image, point-prompt based approach without requiring additional adaptation or selection. In response to this, we revised the related work section to better reflect this aspect. Thank you for bringing this to our attention.
>
> > Superpixel approach
>
> In our experimentation, we observed that introducing negative points into the model's process in some cases, did not lead to accuracy improvements, but rather resulted in performance deterioration. This phenomenon was particularly noticeable in scenarios where the model lacked the ability to differentiate between foreground and background. This observation underscores the significance of foreground-background separation, especially in medical datasets where color clustering may not inherently convey meaningful foreground-background distinctions.
>
> > Larger objects
>
> We appreciate the reviewer's inquiry regarding the performance discrepancy between SAM alone and SAM combined with J (self-similarity maps) on the AMOS22 dataset. The lower resolution of SAM+J, which is 16 times lower than the full image resolution and 4 times lower than the original SAM resolution, could be a contributing factor. In datasets where the shape of individual objects is well-defined, like in AMOS22, the low resolution of SAM+J may not provide any advantage over using SAM with points as prompts. However, in cases where objects are repetitive and their shapes are not well-defined, SAM with the original prompts might perform poorly compared to using the self-similarity map.
>
> > The output of (vanilla) SAM
>
> We apologize for any confusion caused. By stating "Obtained similar findings," we intended to convey that we replicated the performance results reported in the SAM.MD short paper using our framework. As a result, we did not encounter any discrepancies or issues.
>
> > Renaming to soft-Dice loss:
>
> Thank you for the clarification regarding your comment. We appreciate your attention to detail. We revised the equation on page 5.
>
> > Layout of the paper:
>
> In response to the feedback regarding the layout of the manuscript, we have taken into account the suggestion to disregard the “vspace” and have increased the font size of all small text throughout the tables. Additionally, we have considered relocating some tables.

---

### Official Review · Reviewer_k7S5 · 2024-02-29

**Confidence:** 4
**Preliminary Rating:** 3
**Final Rating:** 4

**Summary:**

The paper introduces a fine-tuning approach that utilizes initial masks from self-similarity maps to enhance visual encoding for zero-shot segmentation tasks. Unlike other methods that fine-tune the decoder, this paper proposes fine-tuning the visual encoder. The results indicate that this fine-tuning method surpasses alternatives that only provide pseudo points or masks.

**Strengths:**

1. The paper introduces SAM fine-tuning for segmentation without labels, using only pseudo-labels initiated by SAM.
2. It proposes a novel fine-tuning approach focusing on the visual encoder instead of the commonly targeted decoder in other SAM fine-tuning studies.

**Weaknesses:**

1. The quality of the fine-tuned segmentation results is poor. Figures 1, 2, and 3 show segmentation masks with coarse shapes and boundaries, appearing to lose basic morphological patterns.
2. There is no comparative analysis of segmentation performance between fine-tuning the visual encoder and the mask decoder. The rationale and benefits of the proposed fine-tuning approach are not clearly articulated.
3. The paper does not provide SAM segmentation results with full prompting to establish the upper bound of segmentation tasks.

**Detailed Comments:**

The paper should clarify how fine-tuning improves segmentation performance despite relying solely on imperfect initial masks from self-similarity maps. What additional knowledge does the model acquire during training?

**Justification Of Final Rating:**

The author addressed my concerns during the rebuttal period, leading me to increase the score for the final judgment. The only concern is the quality of the fine-tuned results, which still do not accurately follow the object morphology. However, the general idea of the paper is interesting and promising for future endeavors.

**Justification Of The Preliminary Rating:**

The paper presents an intriguing approach to zero-shot fine-tuning by adjusting the visual encoder within the Self-Attention Model (SAM), while keeping the decoder's weights frozen. However, to fully demonstrate the method's effectiveness and uniqueness, additional comparisons and clarifications are necessary. Consequently, I am assigning a borderline rating to this paper.

**Questions To Address In The Rebuttal:**

Please address the questions in weaknesses and detailed comments.

---

> ### Author Response · Authors · 2024-03-16
>
> We thank the reviewer for the comprehensive feedback.
>
> > Lack of Comparative Analysis
>
> In Appendix B, we included an analysis of the fine-tuning process in comparison to the initial mask $J$. The figure highlights the significance of this procedure in refining the output mask for the majority of the data points.
>
> > Fine-tuning the visual encoder and the mask decoder
>
> We chose not to optimize the decoder because doing so could cause it to simply follow $J$ and converge to it without refining it. By freezing the decoder and solely optimizing the encoder, the output mask cannot follow $J$ precisely and therefore refines it, as demonstrated in Appendix B.
>
> >SAM with a complete prompt:
>
> For each table presented in the results section, we included a reference to the state-of-the-art supervised algorithm. As observed in the AMOS22 dataset, the gap between our method and the supervised algorithm is relatively narrow.

---

### Official Review · Reviewer_f18q · 2024-03-05

**Confidence:** 3
**Preliminary Rating:** 4
**Final Rating:** 4

**Summary:**

The authors propose a method to take advantage of the progresses of fundamental segmentation models in the medical domain, by reducing the number of prompt inputs needed when working on images with multiple foreground segments. To this end, they do a retraining of the encoder of the SAM model to help it overcome the domain shift when working on zero-shot learning, this retraining being based on the cosine similarity between the embeddings of each pair of pixels of the image.

**Strengths:**

- The paper proposed an interesting way to use the similarity of the SAM embeddings of the pixels and the point prompts to facilitate the use of the SAM model.
- The method brings good improvement to a direct use of the SAM model on medical images.
- Rich presentation of qualitative results.
- Opening the source code is a great point for the reproducibility of the method.

**Weaknesses:**

- Confronting the proposed method to other types of data of different scales (ex : CT/PET images, MRI, ...) would give a better understanding of its capabilities.
- A figure explaining the model would be of a great help to understand the process at hand.

**Detailed Comments:**

- The figures' position is counter-intuitive (if it can be changed within the framework), and another round of proofreading could be helpful (doubled words, some equations are not well written ('W times H'), sentence without a verb)

**Justification Of Final Rating:**

Interesting method that seems to bring improvement on some medical segmentation tasks, where we're constrained by the quantity of annotated data available, with few effort to make to define prompts for the model to use. A good effort was made to answer the questions of the reviewers and to complete and clarify the work.

**Justification Of The Preliminary Rating:**

Interesting method that seems to bring improvement on some medical segmentation tasks, where we're constrained by the quantity of annotated data available, with few effort to make to define prompts for the model to use.

**Questions To Address In The Rebuttal:**

I would be interested in seeing if this progress is generalizable on other types of data, maybe on tasks where the background is more variable.

---

> ### Author Response · Authors · 2024-03-16
>
> We thank the reviewer for the comprehensive feedback.
>
> > Comparison with Other Data Types
>
> Following your request, we have incorporated the AMOS22 dataset as a new benchmark that has a different type of data. Our method demonstrates superior performance compared to the other baselines, as evidenced by the results presented in Table 1.

---

### Author Response · Authors · 2024-03-16
**General Response**

We thank the reviewers for their valuable feedback and suggestions. Following the reviews, we have added a benchmark (AMOS22) that is not a cell segmentation task, as well as an additional baseline method. The added results strengthen the utility of our method.

We have uploaded a revised version. A summary of the changes is provided below:
* Following reviewer aU6b, we have added more content to the related work section.
* Following all reviews, the AMOS22 benchmark was added.
* Following reviewer aU6b, a new baseline ‘’Super-cluster’’ was added.
* We have fixed the typos raised by the reviewers
* Following reviewer k7S5, a graph that presents the improvement of the finetune process was added as appendix B.
* Following reviewer aU6b, the dice loss has been renamed to soft-dice loss.

---

### Meta-Review · Area_Chair_wTXy · 2024-04-04

**Recommendation:** Accept (Poster)
**Confidence:** 4

**Metareview:**

This paper presents a zero-shot image segmentation method by fine-tuning SAM and using very sparse prompts.

The paper received rigorous reviews and discussion, and based on the concerns of the reviewers, mostly involving more details/related works, as well as quite a bit of experimental improvement. The original reviews were borderline. Following this, the authors went through careful revision of the work and responded with cleaner experiments and manuscript. Overall, all reviewers were pleased.

Although there were substantial changes from the original manuscript, I think the paper is now in shape to be insightful to the community and should be accepted. As reviewers note, there are still some concerns left, and I encourage the authors to take these into account when they present the work at the conference.

---

### Decision · Program_Chairs · 2024-04-06

Accept (Poster)